# Migration of BPA from Food Packaging and Household Products on the Croatian Market

**DOI:** 10.3390/ijerph20042877

**Published:** 2023-02-07

**Authors:** Adela Krivohlavek, Nataša Mikulec, Maja Budeč, Lidija Barušić, Jasna Bošnir, Sandra Šikić, Ivone Jakasa, Tajana Begović, Rea Janda, Ksenija Vitale

**Affiliations:** 1Teaching Institute for Public Health “Dr. Andrija Štampar”, 10000 Zagreb, Croatia; 2Faculty of Agriculture, University of Zagreb, Svetošimunska Cesta 25, 10000 Zagreb, Croatia; 3Faculty of Food Technology and Biotechnology, University of Zagreb, 10000 Zagreb, Croatia; 4Faculty of Science, University of Zagreb, 10000 Zagreb, Croatia; 5School of Medicine, University of Zagreb, 10000 Zagreb, Croatia

**Keywords:** BPA, household products, food packaging, health, Croatia

## Abstract

BPA is a plasticizer for the production of polycarbonate plastics and epoxy resins and is widely used in the production of household goods, including food packaging. Free BPA is known to migrate from packaging to food, and its uptake has been associated with adverse health effect, particularly the disruption of endocrine activity. The presence and migration of BPA from plastic consumer products are subject to strict regulation in the EU. The aim of this study is to analyse the migration of BPA from different packaging items and household products sold on the Croatian market. To simulate real life exposure, we treated samples with a food simulant. The analytical performance was confirmed with the EU requirements. BPA levels were assessed in 61 samples by HPLC-FLD and the LOQ of the method was 0.005 mg kg^−1^ for the food simulant. These results showed that the levels of BPA that migrated to the food simulant were below LOQ and in accordance with the specific migration limit into food, which was defined as 0.05 mg kg^−1^ for all samples. None of the analysed products presented a health hazard. However, these regulations do not refer to products intended for children’s use, in which BPA is banned. Furthermore, regulations require testing before putting products on the market, and previous research shows that possible BPA migration occurs due to various uses, along with a cumulative effect of exposure from even very small concentrations. Therefore, for accurate BPA consumer exposure evaluation and possible health risks, a comprehensive approach is needed.

## 1. Introduction

Bisphenol A (BPA) or 2,2-Bis(4-hydroxyphenyl) propane (CH_3_)_2_C(C_6_H_4_OH)_2_ is an organic compound with two phenol functional groups. At room temperature, it is a colourless solid with a melting point between 153 °C and 156 °C and a boiling point at 200 °C. BPA is soluble in organic solvents but poorly soluble in water. In its free form, BPA is somewhat lipophilic, with an experimentally determined logP value of 3.32 [1]. According to Cameo Chemicals, BPA is combustible and may form explosive dust clouds [2]. It is used as an intermediate in the production of polycarbonate (PC) plastics and epoxy resins. PC plastics are used to produce various food containers, including bottles, while epoxy resin’s main purpose is the prevention of the direct contact of food with metal walls in food cans [3]. BPA is a known environmental pollutant and endocrine disrupting chemical, and as such is found in the environment in concentrations that cannot be dispersed, diluted, decomposed, recycled, or stored in some harmless form. They have negative effects on the environment and wildlife and often impact human health and well-being [4,5]; therefore, the use of bisphenol A is limited in many countries. At the moment, for the manufacturing of polycarbonate plastics, bisphenol S (BPS) and bisphenol F (BPF) are proposed as safe alternatives. They have already been released into the aquatic environment without previously available information about their potential adverse effects. All endocrine disrupting chemicals mimic or interfere with the normal actions of hormones in the body, affecting essential biological processes, including development, metabolism, and reproduction. A number of studies have shown that BPA causes female and male infertility, precocious puberty, hormone-dependent tumours and several metabolic disorders [6,7,8,9,10]. Humans are exposed to BPA mainly through ingestion by the consumption of food and beverages stored in plastic containers, bottles, and cans [11,12]. BPA has been detected in numerous foods such as fruits and vegetables, cereals, meat, fish and seafood, dairy products, baby food, canned beverages, food and drinking water [13,14,15,16]. Gorecki et al. showed that the BPA content in non-canned meat arises post-mortem, i.e., during food production or from the surrounding environment, which was recently reconfirmed [17,18]. Inhalation and dust ingestions seems unlikely to be an important exposure source [19]. A number of studies have also confirmed that dermal absorption can contribute to exposure to BPA [20,21] through contact with thermal paper in which BPA is added as a printing developer [22], and especially after the use of skin and hand sanitizers containing dermal penetration enhancing chemicals [23]. Interestingly, water disinfection methods can aid in the removal of BPA from drinking water [24,25,26,27].

Due to its adverse health effects, BPA is one of the major concerns of regulatory agencies all around the world. According to the U.S. Environmental Protection Agency [28], the intake limit for BPA is 0.05 mg per kg of body weight (bw) per day, while the European food safety authority (EFSA) and the European Commission (EC) have significantly lowered the tolerable daily intake (TDI) value to 0.004 mg kg^–1^ bw day^–1^. The EC has also set a specific migration limit for BPA in food to 0.5 mg kg^–1^ [12,28,29]. According to the EC, the maximum limit value of BPA in plastic materials and articles intended to come into contact with food is 0.05 mg kg^–1^ (Commission Regulation (EU) No 10/2011) [30]. Croatian normative documents are in accordance with all EU regulations, and in most cases they are simply translated and accepted by Croatian Standards Institute.

Orzel and Swit [31] compared quantitative detection methods based on molecular fluorescence spectroscopy and chromatographic techniques used for the determination of bisphenol compounds and concluded that high performance liquid chromatography with fluorescence detector (HPLC-FLD) result in the highest sensitivity and the lowest limit of detection (LODs) values. Due to similar spectral properties of BPA and BPF, simultaneous detection using the fluorescence approach is impossible [31]. This work supports our previous decision to use HPLC-FLD for the qualification and quantification of BPA migration from different packaging items and household products.

The aim of this study was to analyse the migration of BPA from different packaging items and household products sold on the Croatian market before first use, and whether these results are in compliance with BPA migration limits set in EU Commission Regulation (EU) No 10/2011 [30]. Only BPA migration limits are set, so all our monitoring was done for BPA.

## 2. Materials and Methods

### 2.1. Reagents and Materials

Methanol (gradient grade for LC) was purchased from Supelco (Deventer, Germany). Bisphenol A (BPA) analytical standard was purchased from Dr. Ehrenstorfer (Augsburg, Germany). Glacial acetic acid (p.a.) was purchased from Alkaloid (Skopje, Northern Macedonia). Water (0.055 µS cm^−1^) was obtained from an EVOQUA, BKG Wassertechnik, GMBH (Eningen, Germany) water purification system.

Individual stock solution of BPA was prepared in methanol at a concentration level of 375 µg mL^−1^. Working standard solution of BPA was prepared by appropriate dilution of stock solution of BPA in 3% acetic acid aqueous solution. The calibration solution’s concentration range was 0.005–0.8 mg kg^−1^. A measure of 3% acetic acid in aqueous solution was also used as food simulant for all analyses. All solutions were prepared fresh at the day of analysis. 

### 2.2. Sample Collection

Sample consisted of products intended to come into contact with food that were provided by importing companies, the food industry, and the Croatian Ministry of Health, Zagreb’s City office for Health and Croatian State Inspectorate. In total, 61 samples were collected from 2012 to 2022 and analysed for migration of BPA to aqueous food simulants and in foodstuff according to HRN CEN/TS 13130:2005 norm Part 1 and 13 [32,33]. Samples were stored at room temperature in original delivery package and analysed within a week of arriving at the laboratory. These samples were intended for households (61) and included products from plastic materials or metal and paper product materials lined with a thin layer of plastic material. 

### 2.3. Sample Preparation

Samples were first gently wiped with clean non-linting cloth to remove surface contaminants. Alternatively, samples were washed with ultra-pure water and left at room temperature to dry, then were gently wiped with non-linting cloth. Samples were analysed for the migration of BPA to aqueous food simulants according to HRN CEN/TS 13130:2005 norm Part 1 and 13 [32,33], Commission Regulation (EU) No 10/2011 [30], and No 1245/2020 [34]. The release of BPA was evaluated by filling or soaking each article with prescribed food simulant up to a nominal volume and were left in contact with food simulant for a standardized time and temperature. 

### 2.4. Chromatographic Analysis and Instrumentation

Analysis of BPA was performed in our ISO 17,025 accredited laboratory with in-house developed methods following standard procedure according to HRN EN ISO/IEC 17025:2017 [35]. An Agilent liquid chromatograph Serie 1100 with fluorescence detector (FLD) was used for the analysis of BPA. Analytes were separated on a 250 × 4.6 mm reversed-phase Lichrospher RP-18 5 µm column (Phenomenex, Torrance, CA, USA) at a flow rate of 0.6 mL min^−1^ under isocratic conditions, with mobile phase consisting of 65% methanol in water. The column was thermostated at 30 °C and the injection volume of food simulant was 40 µL. Fluorescence excitation was set to 275 nm and emission to 313 nm as in HRN EN 71-11:2008 [36], which was used for setting chromatographic parameters.

The analysis time was 20 min and calibration curve range and method operating range was 0.005–0.50 mg kg^−1^. Conformation of BPA identity was based on retention time being within ±0.01 min of the standard. The concentration of BPA in the food simulant was based on the calibration curve based on plotting peak area to the BPA concentration. Limit of quantification (LOQ) for BPA in food simulant samples was 0.005 mg kg^–1^. All validation parameters are shown in Table 1.

All validation parameters shown in Table 1 are in accordance with HRN EN 71-11:2008 [36]. Validation parameters are experimentally determined and accepted by Institute internal Standard Operative Procedures (SOPs) for method validation.

## 3. Results

The products tested for BPA migration covered a wide range of various household products used by the general population in everyday life. Only one tested product was intended for industrial use (milking equipment). Although at first glance some products had the same nominal description, such as “baby bottle”, they differed according to the detailed description by the producer when coming with the original packaging of the product. Some products came without a description of the material they were made of or at least coated with; therefore, when information was available on internet it was included in the general information. Some products had information on nominal volume to which they were filled, although actual volume was not known. Regardless, all samples that provided information on material, coating or volume were tested. All results were presented in Table 2. The results showed that all of the analysed products for baby use were below 0.005 mg kg^−1^, and the other analysed products were below the limits of specific migration of 0.05 mg kg^–1^.

As presented in Table 1, all results showed BPA levels were below LOQ (0.005 mg kg^−1^) for food simulant, which were in accordance with the specific migration limit into the food defined as 0.05 mg kg^−1^ for all samples. These samples covered the whole spectrum of household-use products ranging from wrapping materials, containers, and tableware to baby products. Regulations refer to all products except products intended for children use, in which BPA is banned.

Chromatograms of BPA standard solution (a), spiked sample (b) and sample (c) are shown in Figure 1.

As is shown in Figure 1, the chromatograms of BPA standard solution (a) and spiked sample (b) BPA retention time are for 15.5 min. For sample (c) there is no peak on that retention time, and the result for BPA is less than 0.005 mg kg^−1^.

## 4. Discussion

A comprehensive study on the presence of BPA in items of general use on the Croatian market and its migration to food simulant has not been conducted until the present study. A search of the Science Direct database by keywords BPA and Croatia yielded only two studies from Croatia, both regarding the toxic and reproductive impact of BPA on soil invertebrates [39] and sea urchins [40]. As for other endocrine disruptors from food packaging and plastic household items, only one research on phthalates has been conducted, where level and rate of migration of phthalates to model solution was determined. Results indicated a low exposure to the phthalates through examined plastic items, but authors argued that humans could be exposed to phthalates from more than one source (water, food, medical products), receiving doses that could accumulate in the body. As a conclusion, the authors stated that maximal levels of phthalates in the various environmental sources should be determined [41]. 

Croatia is a small country with a population of around 3.9 million (2021) and with a limited market share. Therefore, once imported and approved, some products are on the market for long period of time; therefore, only a limited number of different samples could be collected and analysed. 

Our samples were mostly made of polycarbonate or epoxy resins, which are known for their use of BPA in their production as well as other alternative materials, e.g., polypropylene, which is often marketed as “BPA free” material. Levels of BPA in the food simulant, regardless of the item’s type of material or temperature and duration of contact with the food simulant in the migration tests, were below the limit of quantification of the method (0.005 mg kg^−1^) for all samples tested (Table 2), and therefore below the specific migration limit for foodstuff or food simulant of 0.05 mg kg^−1^. In this study, we limited analysis to only BPA and not its substitutes due to the EU Directives requirements. 

At first glance these results indicate that items put on the Croatian market comply with the maximum allowed concentrations and do not present a health hazard, regardless of the type of material of which they are made. We have to highlight, though, that the items were tested for BPA migration before their first use, and that for the migration test, only aqueous 3% acetic acid as simulant was used. Park et al. showed that the migration of BPA to food simulant was highest in aqueous 50% ethanol at a temperature of 70 °C in comparison to aqueous 4% acetic acid, water or *n*-heptane; however, it was still below current SML (specific migration limit) [42]. Biedermann-Brem et al. found that the migration levels of BPA from the walls of polycarbonate bottles, even in extreme situations such as high temperatures in combination with washing scenarios with detergent and use of strong alkali (which contribute to the degradation of polycarbonate material), do not result in BPA contamination near the level corresponding to the TDI (total daily intake) [43]. 

Baby bottles and cups are under special concern because they are used in a period of intense growth and development when the body mass of a baby or infant is still relatively low. In our study, baby bottles were made mostly of PP and declared BPA free and had a small number of PCs and PES. The level of the migration of BPA to the food simulant was below the limit of quantification, regardless of material. The same types of bottles were tested by Russo et al., 2018 [44], showing that after typical daily use and heating to 40 °C and 80 °C, BPA was detected even though the bottles were marketed as BPA free. The highest concentrations were detected in PP bottles heated in a microwave oven at 40 °C. Similar findings were published by Ali et al. and Osman et al., 2018, who found that the BPA concentration in baby formula was almost 10 times higher after the bottle was used 100 times and heated at 90 °C [45,46].

An analysis of water bottles for multiple uses and made from both plastic and aluminium with specific coating on the Slovenian market [47] revealed similar results. On the Austrian market, BPA-free multiuse water bottles were tested for migration into water and artificial saliva at 60 °C and 90 °C. BPA was detected in all samples [48]. Studies on polycarbonate bottles showed that increased temperature and repeated use caused an increased release of BPA to water [3], but the concentration in the water was still low enough not to exceed the TDI [49,50]. Additionally, bottles for soft drinks were also tested [51] and the highest concentrations were detected in drinks packaged in glass bottles. These results indicate that contamination with BPA might come from sources other than bottle material, i.e., during food processing or from the environment.

However, there is a reasonable concern regarding how precise or realistic food simulants simulate real food mixtures, especially in the various conditions of temperature, light, pH, storage time or contact area [52,53]. Some authors speculate that food simulants could be useful in some cases, but the final risk assessment can be performed only using data from BPA migration analyses with real food. There are very few comparative studies that compared the migration of BPA to food simulant and real foodstuff under the identical conditions. One example is a study by Česen et al. [54], who found that the BPA levels in honey were almost two times higher than in the food simulant. Contrary to this study, Munguia-Lopez et al. [55] found that the migration level of BPA was overestimated when using a food simulant in comparison to its migration to real food (in this case jalapeño peppers), although in both cases levels were below current SML.

Although our current study, as well as some other studies, show that migration levels of BPA to food simulants are below current SML, the question of the frequency of use of various products, cumulative effect and variety of exposure arises. A number of authors [45,46,48,51,56,57] suggest that humans are almost ubiquitously exposed to BPA and food contamination that might come from different sources, such as contact material during cleaning and contact material during preparation, along with different packaging materials that might have a cumulative effect. Ginter-Krmarzyk et al. [52] demonstrated that even though the individual concentrations of BPA in bottled water might be low (ng L^−1^) at 0.6 mg kg^−1^ per body weight, the cumulative daily dose in the body may be much higher than the quoted concentrations due to the number of products containing BPA. It should be kept in mind that food processing technology itself could be a source of BPA contamination in food and water [17,58,59,60]. In today’s world, where it is almost impossible to find food that is not processed at least to some extent, this is one more reason for a comprehensive approach in BPA exposure assessment.

## 5. Conclusions

The samples analysed for BPA migration to food simulant under different temperatures and storage durations were below the limit of quantification of the method (0.005 mg kg^−1^) for all samples tested, complying with the requirements of EU regulations and directives. However, it must be stressed that these regulations do not refer to products that are intended for children use, and that regulations require testing only before putting products on the market, while previous research results show that possible BPA migration may occur later due to the various uses (temperature, washing, pH). Also, research has demonstrated possible cumulative effects of exposure to even small concentrations of BPA. Our findings support the idea of quicker regulations and directive changes according to the newest scientific findings. Therefore, for accurate BPA consumer exposure evaluation and possible health risks, a comprehensive approach is needed.

## Figures and Tables

**Figure 1 ijerph-20-02877-f001:**
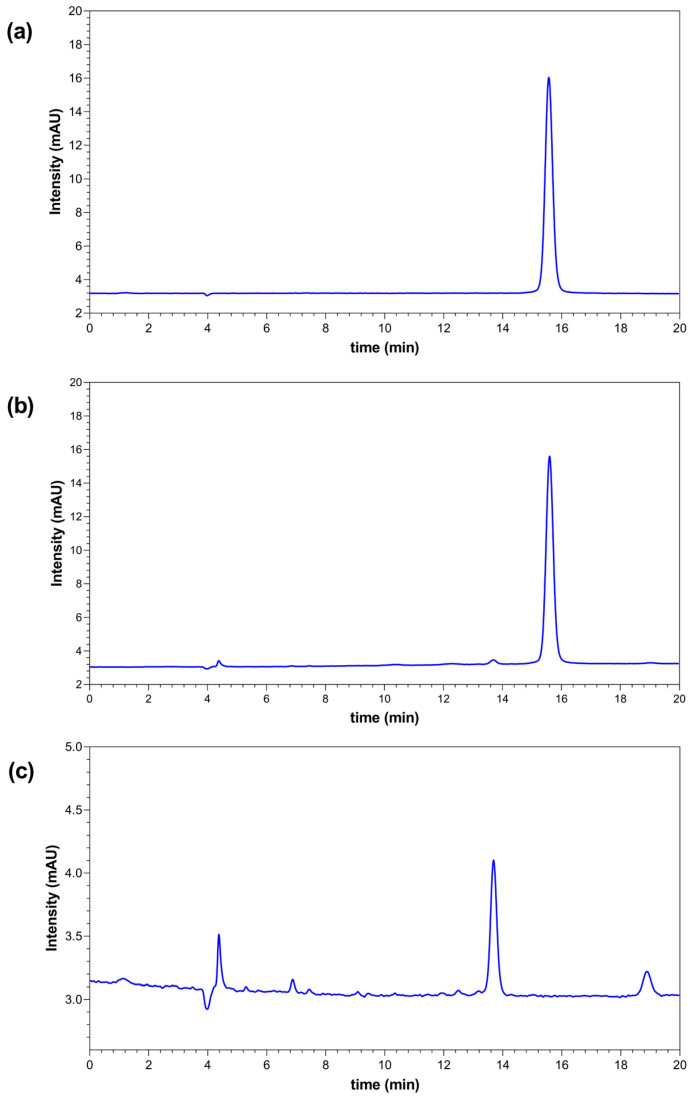
HPLC-FLD chromatograms of BPA standard solution (**a**), spiked sample (**b**) and sample (**c**).

**Table 1 ijerph-20-02877-t001:** Validation parameters for determination of BPA in food packaging and household products.

Validation Parameters	Experimental Data	Acceptance Criteria of Validation Parameters
Linearity	k = 0.99998	k ≥ 0.999
Recovery	95.2%	100 ± 5%
Precision:		
Repeatability of sample preparation (RSD)	4.00%	RSD * ≤ 5%
Intermediate precision (RSD)	4.56%	RSD * ≤ 5%
Limit of quantification	0.005 mg kg^−1^	10× < MAC **
Limit of detection	0.002 mg kg^−1^	25× < MAC **

* RSD relative standard deviation; ** MAC maximum allowed concentration.

**Table 2 ijerph-20-02877-t002:** Description of the samples, migration time and temperature, and results of BPA migration test in materials and objects that came into contact with food according to Regulation (EC) No 1935/2004 [37], artificial mass. Commission Implementing Regulation (EU) No 321/2011 [38] specifically deals with Bisphenol A in articles for children, and Commission Regulation (EU) No 213/2018 [29] for BPA used in varnishes and coatings that come into contact with food (SLM < 0.05 mg kg^–1^) already implemented in Commission Regulation (EU) No 10/2011 [30].

Material/Coating/Declaration Claim	Sample Description	Migration Time/Temp	BPA Migration (mg/kg)
Tritan coplyester and PP, BPA and DEHP free	Shaker shieldmixer (700 mL)	24 h/40 °C	<LOQ
N/A	Breast milk bag (100 mL)	10 days/20 °C	<LOQ
Inner side of lid with PE phenol and organosol	Metal lid with gasket (100) (ID: 4 cm)	10 days/40 °C	<LOQ
coating, BPA n.i., PVC coating of gasket	Metal lid 2 (100) (ID: 4 cm)	10 days/40 °C	<LOQ
epoxy and PE coating	Bottle for water (600 mL)	24 h/40 °C	<LOQ
Cellulose bioplastics	Cup (200 mL)	24 h/40 °C	<LOQ
N/A	Welded tin can (100 mL)	10 days/40 °C	<LOQ
N/A	Ceramic mug with silicone lid (100 mL)	2 h/70 °C	<LOQ
PVC coating of gasket	Metal lid with gasket 1(100 mL)	10 days/40 °C	<LOQ
PVC coating of gasket	Metal lid with gasket 2 (100 mL)	10 days/40 °C	<LOQ
Polymer coating (unknown material)	Beer tin can 1 (500 mL)	10 days/40 °C	<LOQ
N/A	Pizza stopper (100 mL)	2 h/70 °C	<LOQ
PC and silicone pacifier	Baby bottle with 14 (270 mL)	24 h/40 °C	<LOQ
Melamine	Cup (225 mL)	2 h/70 °C	<LOQ
PET	Bottle preform (1.5 L)	10 days/40 °C	<LOQ
Melamine	Cup (* *nominal volume*)	2 h/70 °C	<LOQ
N/A	Cup with straw (* *nominal volume*)	2 h/70 °C	<LOQ
N/A	Beer tin can 2 (500 mL)	10 days/40 °C	<LOQ
Epoxy coating	Colander (1000 mL)	1 h/100 °C	<LOQ
Tritan TM copolyester, BPA free	Bottle 1 (600 mL)	24 h/40 °C	<LOQ
Tritan TM copolyester, BPA free	Bottle 2 (700 mL)	24 h/40 °C	<LOQ
Tritan TM copolyester, BPA free	Bottle 3 (800 mL)	24 h/40 °C	<LOQ
PP	Baby bottle (* *nominal volume*)	24 h/40 °C	<LOQ
N/A	Paper-based straw (0.7 × 240 mm) 100 ml	1 h/20 °C	<LOQ
PP, BPA free	Baby bottle 10 (150 mL)	24 h/40 °C	<LOQ
PP, BPA free	Baby bottle 11 (125 mL)	24 h/40 °C	<LOQ
PP, BPA free	Baby bottle 12 (150 mL)	24 h/40 °C	<LOQ
PP, BPA free	Baby bottle 13 (260 mL)	24 h/40 °C	<LOQ
PC	Baby bottle 1 (150 mL)	24 h/40 °C	<LOQ
PP BPA free	Baby bottle 2 (260 mL)	24 h/40 °C	<LOQ
PC	Baby bottle 3 (270 mL)	24 h/40 °C	<LOQ
PC	Baby bottle 4 (300 mL)	24 h/40 °C	<LOQ
PP BPA free	Baby bottle 5 (330 mL)	24 h/40 °C	<LOQ
PES	Baby bottle 6 (125 mL)	24 h/40 °C	<LOQ
N/A	Baby bottle 7 (* *nominal volume*)	24 h/40 °C	<LOQ
PP	Baby bottle 8 (* *nominal volume*)	24 h/40 °C	<LOQ
Melamine	Plastic plate 1 (* *nominal volume*)	24 h/40 °C	<LOQ
N/A	Plastic cup 2 (270 mL)	2 h/70 °C	<LOQ
N/A	Cake mould 1 (1000 mL)	2 h/100 °C	<LOQ
N/A	Cake mould 2 (1000 mL)	2 h/100 °C	<LOQ
N/A, BPA free	Lunch box and drinking bottle (* *nominal volume*)	24 h/40 °C	<LOQ
N/A	Sauce pan with non-sticky layer (ID: 24 cm 1000 mL)	2 h/100 °C	<LOQ
PELD	White foil (166.7 mL/1 dm^2^)	10 days/40 °C	<LOQ
N/A	Bags for sauerkraut (1000 mL)	10 days/40 °C	<LOQ
PP	Plastic bin (1.2 L)	10 days/40 °C	<LOQ
PC	Blue gallon for water (18.9 L)	10 days/40 °C	<LOQ
N/A	Bottle 4 (500 mL)	10 days/40 °C	<LOQ
PC	Gallon for water (18.9 L)	10 days/40 °C	<LOQ
N/A	Tin can with lid (100 mL)	10 days/40 °C	<LOQ
OPA/PELD	Foil	10 days/60 °C	<LOQ
PP	Bin with cover (1.200 L)	10 days/40 °C	<LOQ
N/A	Milking equipment-silicone milk tubing (100 mL)	2 h/40 °C	<LOQ
HDPE	Multilayer/multi-material cup (7 mL)	10 days/60 °C	<LOQ
N/A	Primary bottle with pump (100 mL)	10 days/40 °C	<LOQ
PP	Multilayer cup (11 mL)	10 days/60 °C	<LOQ
N/A	Metallic dish 1 with clear lacquer (1000 mL)	10 days/40 °C	<LOQ
N/A	Coffee tin can with clear lacquer (1000 mL)	10 days/40 °C	<LOQ
N/A	Metallic dish 1 with clear lacquer (1000 mL)	10 days/40 °C	<LOQ
Aluminium pigment polyester coating, BPA n.i.	Lithographed towed can from aluminium tin 1 (50 mL)	10 days/40 °C	<LOQ
Aluminium pigment epoxy phenolic coating, BPA n.i.	Lithographed towed can from aluminium tin 2 (100 mL)	10 days/40 °C	<LOQ
Aluminium pigment epoxy phenolic coating, BPA n.i.	Lithographed towed can from aluminium tin 3 (100 mL)	10 days/40 °C	<LOQ
N/A	Candy box (≈150 mL)	10 days/40 °C	<LOQ

PE → polyethylene, PC → polycarbonate, PES → polyethersulphone, PP → polypropylene, PA → polyamide, PVC → polyvinylchloride, HDPE (PEHD) → high density polyethylene, PELD → low density polyethylene, BPA n.i. → BPA non-intent, DEHP → Bis(2-ethyhexyl) phthalate, PET → polyethylene terephthalate, N/A → not applicable, type of polymeric coating unknown, * *nominal volume* → article was filled to the nominal volume (actual volume not known). LOQ → limit of quantification.

## Data Availability

Supporting data are available at Teaching Institute for Public Health “Dr. Andrija Štampar”, Mirogojska cesta 16, 10000 Zagreb, Croatia.

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
