# Peer review of "Migration of BPA from Food Packaging and Household Products on the Croatian Market"

_ijerph, 2023, doi:10.3390/ijerph20042877_

Round 1
Reviewer 1 Report
The present manuscript entitled "Migration of BPA from food packaging and household products on Croatian market" by Adela Krivohlavek, Nataša Mikulec, Maja Budeč, Lidija Barušić, Jasna Bošnir, Sandra Šikić, Ivone Jakaša, Tajana Begović, Rea Janda, and Ksenija Vitale (ijerph-2174808) is written correctly and has a good structure; moreover, it has all the necessary parts. The article is interesting from an analytical, environmental, and health point of view; therefore, it should interest the reader. I proposed improvements in the method description and with a presentation of figures. The paper meets the International Journal of Environmental Research and Public Health requirements, and I recommend the article for publication in the International Journal of Environmental Research and Public Health following the common editing stage. My current decision is a major revision. More specific comments and observations are presented below.
1. Introduction. Please, add a paragraph about the analytical techniques used to test this analyte.
2. Page 2, line 84. Please replace "Bisphenol" with "Bisphenol A".
3. Page 3, lines 102 and 112. Table 2 appears in the text before Table 1. They should appear sequentially. Please renumber.
4. Page 3, lines 113-121. How was the method optimized? Were experimental design methods used? How are the excitation and emission wavelengths chosen?
5. How were the validation parameters determined?
6. Table 1. What does time/temp mean? This is not clear regarding method validation. Commas should be replaced with dots. RSD expressed as a percentage is the coefficient of variation (CV). Use "LOD" and "LOQ" for recording. What does "Information" mean? The table should be better commented in the text.
7. Table 2. Please replace "LoQ" with "LOQ".
8. Figure 1. Figures should be enlarged for better visibility. Please add descriptions of the axes with the unit. Is it possible to export the data and prepare the chromatograms in better quality?
9. Were interferences checked? What can be done in the event of strong interference effects? How would you deal with them? What types of interference effects could occur?
10. How many times were the measurements repeated?
11. Does the conducted studies have disadvantages?
12. Conclusions. Please clearly highlight the most important advantage.
13. Appropriate tools should be used to best characterize the method when developing a new approach (e.g., AGREE- Analytical GREEnness Metric Approach or RGB model)
14. References. Please check with the journal's requirements. The following work may also be cited: Int. J. Mol. Sci. 2021, 22(19), 10569; https://doi.org/10.3390/ijms221910569 .
I hope that the comments presented will help improve the article.
Author Response
Thank you for considering our paper for publication in your journal, and constructive comments from reviewers. Since two reviewers asked for minor and moderate editing of English language and style, we had our manuscript checked and all changes are visible. Both grammar and style are corrected, so we believe that now manuscript is easier to read and that we accomplished better flow of the text.
At this occasion we would like to thank reviewers for their effort in helping us present our work in best possible way.
We accepted all remarks and added what was required.
Below are original comments and our answers:
Reviewer 1.
- Please, add a paragraph about the analytical techniques used to test this analyte.
Response: We added a paragraph about the analytical techniques.
- Page 2, line 84. Please replace "Bisphenol" with "Bisphenol A".
Response: Yes, we made change.
- Page 3, lines 102 and 112. Table 2 appears in the text before Table 1. They should appear sequentially. Please renumber.
Response: Yes, we made change, and reorganized text mentioning tables.
- Page 3, lines 113-121. How was the method optimized? Were experimental design methods used? How are the excitation and emission wavelengths chosen?
Response: Thank You for this remark. In text we clarified how designed methods including excitation and emission wavelength are chosen. We added extra reference (HRN EN 71-11:2008 Safety of toys - Part 11: Organic chemical compounds - Test methods (EN 71-11:2005))
- How were the validation parameters determined?
Response: All validation parameters shown in Table 1 are in accordance with HRN EN 71-11:2008 [36]. Validation parameters are experimentally determined and accepted by Institute Internal Standard operative procedures for method validation.
- Table 1. What does time/temp mean? This is not clear regarding method validation. Commas should be replaced with dots. RSD expressed as a percentage is the coefficient of variation (CV). Use "LOD" and "LOQ" for recording. What does "Information" mean? The table should be better commented in the text.
Response: Thank You for indicating this vagueness. We deleted Time/temperatures as redundancies; changed commas to dots; We agree that RSD is CV, but we decided to use RSD. LOQ is 10 times less than MAC value. LOD is 25 times less than MAC value. All added in Table 1.
- Table 2. Please replace "LoQ" with "LOQ".
Response: Yes, we made the change.
- Figure 1. Figures should be enlarged for better visibility. Please add descriptions of the axes with the unit. Is it possible to export the data and prepare the chromatograms in better quality?
Response: We replaced existing Figures with graphical better ones and added descriptions of axes with units
- Were interferences checked? What can be done in the event of strong interference effects? How would you deal with them? What types of interference effects could occur?
Response: Yes. The selectivity of the method is checked by comparing the chromatograms of the sample and the standard. By comparison of the chromatograms of the sample without BPA and the chromatogram of standard, it is established that there are no obstacles to the quantification of the desired analyte.
- How many times were the measurements repeated?
Response: For the validation purposes 6 replicates were conducted. Each 10th measured sample were analysed in duplicate. For each series recovery test were done. With the fact that the samples are recorded in parallel, it means that they are recorded twice from the same vial - this is the case with all samples. For validation parameters, everything is done 6x, while LOQ and LOD are taken from 10 or 12 injections.
- Does the conducted studies have disadvantages?
Response: Conducted study does not have disadvantages in methodological terms. It is a standard procedure that must be followed. In our discussion, we argue that this procedure itself could be reconsidered due to the growing knowledge on BPA.
- Conclusions. Please clearly highlight the most important advantage.
Response: Yes, we made changes in the conclusion stating demonstrating that
- Appropriate tools should be used to best characterize the method when developing a new approach (e.g., AGREE- Analytical GREEnness Metric Approach or RGB model)
Response: Yes, we completely agree that new methods should consider environmental aspects as stated in medical sciences “do not harm”. As we mentioned before we were obliged to use standard methods, but our institution is dedicated to implement principles of green laboratories and green chemistry in time to come.
- Please check with the journal's requirements. The following work may also be cited: Int. J. Mol. Sci. 2021, 22(19), 10569; https://doi.org/10.3390/ijms221910569
Response: Thank you for the suggestion, we read the article dealing with the detection methods, and added paragraph in the introduction about used techniques. For better perspective we cited proposed article.

Reviewer 2 Report
Krivohlavek et al. report a mini review about the BPA migration from food packaging and household products on Croatian market and the main of the study has not been stated clearly. there is no supporting information about their present study in the ''Abstract''. However, the study can be published in the International Journal of Environmental Research and Public Health after minor revisions.
1 The aim of the study should be stated in the ‘’Abstract’’ and ‘’Introduction’’.
2 Abbreviations should be given where the words appear for the first time and used after that.
3 Line 155: Table 2 or Table 1?
4 Figure 1 should be replaced with high-resolution and enlarged images.
5 No reference should be cited in the ‘’conclusion’’ and the new achievement of this study should be stated.
6 The language of the manuscript should be edited and simplified.
Author Response
Thank you for considering our paper for publication in your journal, and constructive comments from reviewers. Since two reviewers asked for minor and moderate editing of English language and style, we had our manuscript checked and all changes are visible. Both grammar and style are corrected, so we believe that now manuscript is easier to read and that we accomplished better flow of the text.
At this occasion we would like to thank reviewers for their effort in helping us present our work in best possible way. We accepted all remarks and added what was required.
Below are original comments and our answers:
Reviewer 2.
- The aim of the study should be stated in the ‘’Abstract’’ and ‘’Introduction’’.
Response: Yes, we stated aim both in Abstract and Introduction
- Abbreviations should be given where the words appear for the first time and used after that.
Response: Yes, abbrevaitions are given where needed.
- Line 155: Table 2 or Table 1?
Response: Thank you for the comment, we clarified order of citing tables.
- Figure 1 should be replaced with high-resolution and enlarged images.
Response: Yes, we replaced existing figures with graphicaly better ones.
- No reference should be cited in the ‘’conclusion’’ and the new achievement of this study should be stated.
Response: Thank you for this remark, by mistake references remained in the text. We reorganized text in the Conclusion for better clarity. We added achievment as well.
- The language of the manuscript should be edited and simplified.
Response: Yes, text was edited by translator at our institution.

Reviewer 3 Report
In the article of Krivohlavek et al. entitled „ Migration of BPA from food packaging and household products on Croatian market” Authors analyzed the level of BPA migration from packaging to an aqueous food simulant solution. In each case, the concentration of BPA was below the LOQ indicated in the EU regulations. In this case, the results alone mean little.
The authors rightly noted that it is important to know the effects of cumulative exposure to small doses of BPA:
1) so I wonder why the analyzes were not performed multiple times on each of the tested items, and then single results, for example for items that are often used one after the other such as “Breast milk bag” and “Baby bottle”, have not been aggregated? Then it would be possible to test multiple doses of BPA from products that have been "aged" by running the migration process multiple times.
Nevertheless, the overall impression is that the obtained results seem to be reliable and paper is well written, with proper use of citations and in my opinion is suitable for publication in International Journal of Environmental Research and Public Health ,however, data in current form are of low significance
Author Response
Thank you for considering our paper for publication in your journal, and constructive comments from reviewers. Since two reviewers asked for minor and moderate editing of English language and style, we had our manuscript checked and all changes are visible. Both grammar and style are corrected, so we believe that now manuscript is easier to read and that we accomplished better flow of the text.
At this occasion we would like to thank reviewers for their effort in helping us present our work in best possible way.
We accepted all remarks and added what was required.
Below are original comments and our answers:
Reviewer 3.
- so I wonder why the analyzes were not performed multiple times on each of the tested items, and then single results, for example for items that are often used one after the other such as “Breast milk bag” and “Baby bottle”, have not been aggregated? Then it would be possible to test multiple doses of BPA from products that have been "aged" by running the migration process multiple times.
Response: Thank you for the remark, the analyses we performed were obligated analyses prior to entering Croatian market. BPA detection is done only once using standardized method. Our intention was to show, that even when detected concentrations were below requirements, it is still questionable if these products are long term safe. Therefore, our main conclusion and recommendation is to think about quicker change of EU requirements, regulations and directives to be in accordance with newest scientific findings. Single analyses are sufficient at the moment, and only data required by the law, but we suggest that, this is not enough, and that comprehensive (more testing, cumulative effect analyzing etc) approach is needed if we want to be sure of the product quality, particularly in the case of baby products.
As for comment on data significance we strongly believe that data are significant from the point of view of first-time analysis of BPA in Croatia.
Once again, we would like to thank reviewers for valuable comments, that improved our manuscript, and we strongly believe that this new version will be suitable for publishing. We checked references as well and made small corrections.
We hope that we covered all suggested issues in the manuscript and that it will be suitable for publication.
Thank you in advance and best regards,
Nataša Mikulec

Round 2
Reviewer 1 Report
Dear Authors,
Thank you for your meticulous consideration of my comments. The paper has improved substantially and, in my opinion, is suitable for publication.